# Structural and Physicochemical Characteristics of Rice Bran Dietary Fiber by Cellulase and High-Pressure Homogenization

**Fengying Xie** [1,2,†]**, Tian Zhao** [1,†]**, Hongchen Wan** [1]**, Miao Li** [1]**, Lina Sun** [1]**, Zhongjiang Wang** [1,*] **and Shuang Zhang** [1,*]

1    College of Food Science, Northeast Agricultural University, 600 Changjiang Road, Harbin 150030, China; spxfy@163.com (F.X.); ztian597@163.com (T.Z.); 15070927087@163.com (H.W.); li_miao216@163.com (M.L.); minzh@163.com (L.S.)
2    Harbin Food Industry Research Institute, Harbin 150028, China
*    Correspondence: wzjname@126.com (Z.W.); szhang@neau.edu.cn (S.Z.)
†    These authors contributed equally to this work.

**Abstract:** The present paper aims to study the effect of cellulase hydrolysis and high-pressure homogenization on the structural and physicochemical properties of rice bran dietary fiber (RB-DF). Scanning electron microscopy showed that cellulase treatment led to the formation of a porous structure on RB-DF surface. High-pressure homogenization affected the laminated microstructure of RB-DF, leading to the formation of an irregular and loose surface structure. X-ray diffraction demonstrated that joint processing destroyed the amorphous hemicellulose and cellulose regions, and changed the crystallinity of RB-DF, albeit with a minor impact on the crystalline region of cellulose. Fourier transform infrared spectroscopy indicated that combined processing promoted dissociation of some glycosidic bonds in fiber structure, exposing the hydroxyl groups in cellulose, thus improving their ability to bind water molecules. Thermogravimetric analysis showed a significant decrease in the thermal decomposition temperature of RB-DF ($p < 0.05$) as well as a decrease in thermal stability after combined processing. Cellulase hydrolysis and high-pressure homogenization treatment did not improve their oil holding capacity, but significantly increased water holding capacity, swelling capability, and cation exchange capacity of RB-DF. Thus, enzymatic hydrolysis and high-pressure homogenization treatment can change the structure of RB-DF, exposing a large number of hydrophilic groups and enhancing hydration, obtaining uniform RB-DF particle.

**Keywords:** rice bran; enzymatic hydrolysis; cellulase; high-pressure homogenization

## 1. Introduction

Dietary fiber (DF) is the general term for polysaccharide and lignin that cannot be digested and absorbed by biological enzymes in human small intestine [1]. As the seventh most important nutrients for organisms [2], DF is an important component of healthy diet, offering various physiological benefits, including body weight control, serum lipid and cholesterol reduction, controlled postprandial glucose responses, and colon cancer prevention [3].

Rice bran is the outer layer of rice derived as a by-product of milling process and is an ideal source of DF [4,5]. However, the major components of rice bran DF (RB-DF) are insoluble, whereas soluble dietary fiber is of greater benefit to human health [6]. Thus, modification of the insoluble DF to increase its solubility and water holding capacity has become an important research direction for functional properties improvement of RB-DF in food industry. Insoluble DF modification methods include chemical, biological, and physical routes [7]. Chemical methods lead to low yields of soluble

DF, in which harmful chemical groups being easily introduced [8,9]. Biological methods are mild, yet the enzymes used for catalysis are expensive. Thus, the focus is now on physical methods, including high-pressure homogenization (HPH). HPH is a refinement and dispersion technology that can be used for liquid–liquid and solid–liquid systems [10]. The high pressures used lead to a series of changes in the physical, chemical, and structural properties of food products, effectively increasing the soluble fraction of dietary fiber [11,12]. However, few studies have assessed the effect of HPH on the functionality or structure of DF [13,14]. Insoluble DF has a compact polymeric structure that is formed by stiff cellulose and flexible hemicellulose components, which makes it a highly tough structure that is difficult to disintegrate. Breaking the dense structure of insoluble DF is difficult to achieve using a single HPH process, but the use of enzymatic hydrolysis allows the lignocellulose to be cleaved into partial monomer units [15,16]. Additionally, supplementing DF into food requires a substantial understanding of its chemical structure, since the interactions between DF and other ingredients can considerably alter the microstructure and properties of the final products.

Herein, cellulase enzymatic hydrolysis and HPH were combined to treat RB-DF. The characterization and physicochemical properties of RB-DF were determined at different treatment stages. This research should contribute towards the development of a novel and effective method for the modification and application of DF.

## 2. Materials and Methods

### 2.1. Materials

Brown rice was purchased from Wugu Xinhe Agricultural Development Co., Ltd. (Heilongjiang, China). Crude protein, fat, carbohydrate, dietary fiber of brown rice were 7.9 g/100 g, 3.1 g/100 g, 67.6 g/100 g and 5.3 g/100 g respectively. Alkaline protease (enzyme activity $\geq$200 U/mg), $\alpha$-amylase (enzyme activity $\geq$50 U/mg), and cellulase (enzyme activity $\geq$50 U/mg) were obtained from Yuanye Biological Technology Co., Ltd. (Shanghai, China). All other chemicals were of analytical grade.

### 2.2. Rice bran Preparation

Rice bran from brown rice was obtained by polishing the rice using a laboratory rice mill (LTJM-2099, Tuopu Yunnong Technology Co. Ltd., Zhejiang, China) at 2200 rmp for 90 min. The bran samples were stored at 0 °C until further analysis.

### 2.3. Purification of DF

The isolation and purification of DF were performed as described by [17]. The isolation and purification of DF were performed as follows: Rice bran (500 g) soaked in deionized water (1 L, 25 °C, 30 min, 3 times) was centrifuged at 3524 g for 10 min. The insoluble part was dried at 50 °C for 12 h in an air-drying oven, followed by grinding and sieving through a 250 mm mesh and degreasing by washing three times with petroleum ether at a ratio of 1:3. Defatted rice bran (D-RB) mixed (1:6, *w/v*) with 0.2 M phosphate buffer solution (PBS, pH 6.5) was treated with $\alpha$-amylase 24 U/g (D-RB) at 60 °C for 2 h to remove all starch. The slurry was adjusted to pH 10 and 60 U/g (D-RB) alkaline protease was added (2.0 g/100 mL, *w/v*) at 45 °C for 2 h to remove all proteins. The mixture was heated in a boiling water bath for 10 min, and then centrifuged at 3524 g for 10 min. The RB-DF was retained without the supernatant.

### 2.4. Cellulase Enzymatic Treatments

DF mixed (1:6, *w/v*) with 0.2 M acetic acid-sodium acetate buffer (pH 5.5) was treated with 24 U/g (DF) cellulase (2.0 g/100 mL, *w/v*) at 55 °C for 2 h. The treated slurry was centrifuged at 3524 g for 10 min after inactivation of enzyme in boiling water bath for 10 min. The insoluble part was washed thrice with deionized water and centrifuged (3524 g, 10 min).

### 2.5. High-Pressure Homogenization (HPH)

The cellulase enzymatically digested sample (5.0 g) hydrated in deionized water (100 mL) in a plastic bottle (200 mL) was subjected to HPH treatment (FB-110T, Reed Machinery Equipment Engineering Co. Ltd., Shanghai China) for three loop processing with each 30 s at 25 °C using treatment pressure of 120 MPa. Finally, the obtained D-RB, RB-DF, cellulase enzymatic treatment RB-DF (EH-DF), and HPH treatment RB-DF (HPH-DF) were freeze-dried at −50 °C for 24 h, vacuum degree 0.969 mbar using a Freeze dryer (LGJ-10N Sihao Technology Co., Ltd. Hubei, China), and the obtained powders were collected for further analysis.

### 2.6. Scanning Electron Microscopy (SEM)

The surface and microstructure of D-RB, RB-DF, EH-DF, and HPH-DF were observed using a scanning electron microscope (S-3400, Hitachi, Ltd., Tokyo, Japan). Freeze-dried samples were mounted on an aluminum stub with double-sided stick tape, coated with gold, and then examined at an accelerating voltage of 5 kV, according to the method of [18]. Representative images were taken at $500\times$ magnification.

### 2.7. X-Ray Diffraction (XRD)

XRD analysis and crystallinity assessment of D-RB, RB-DF, EH-DF, and HPH-DF were performed as described by [19] with slight modifications. The XRD patterns were obtained using a diffractometer (TTRAX3, theta-theta gonio, USA) with Cu K$\alpha$ radiation ($\lambda$ = 0.15418 nm) over a $2\theta$ range of 5–60°, step of 0.02°/step, and incident current of 150 mA. The degree of crystallinity was determined by calculating area under the curve using Peak Fit v4.12 according to the following equation:

$$D_c(\%) = \frac{A_C}{A_c + A_a} \times 100 \tag{1}$$

where $D_c$ is the degree of crystallinity, $A_c$ is crystallized area, and $A_a$ is amorphous area on the X-ray diffractogram.

### 2.8. Fourier Transform Infrared (FT-IR) Spectroscopy

The structural changes of D-RB, RB-DF, EH-DF, and HPH-DF were measured with a FT-IR spectrometer (MAGNA-IR 560, Nicoet, Ltd., USA), as previously described [20]. Freeze-dried samples (2 mg) were ground with KBr (200 mg, spectroscopic grade), sheeted to one slice, and scanned with a blank KBr background. The FT-IR spectra in a resolution range from 400 to 4000 cm$^{-1}$. A total of 32 scans were collected.

### 2.9. Thermogravimetric Analysis (TGA)

The thermal stability of D-RB, RB-DF, EH-DF, and HPH-DF was measured with a TGA spectrometer (DTG-60, Shimadzu, Ltd., Japan) in a $N_2$ atmosphere. The sample pans were heated from 35 °C to 550 °C, at a rate of 15 °C /min.

### 2.10. Physicochemical Properties

2.10.1. Water and Oil Holding Capacities

The water and oil holding capacities were determined according to the method described by [21], and calculated as follows:

$$WHC(g/g) = \frac{W_1 - W_0}{W_0} \tag{2}$$

$$OHC(g/g) = \frac{W_1 - W_0}{W_0} \tag{3}$$

where $W_0$ and $W_1$ are the weight of the sample before and after oil adsorption, respectively.

### 2.10.2. Swelling Capacity

The swelling capacity was measured according to the method described in Reference [22], and calculated as follows:

$$SC(mL/g) = \frac{V_1 - V_0}{W} \tag{4}$$

where $V_1$ and $V_2$ are the volume of the sample before and after water adsorption, respectively, and W is the weight of the sample.

### 2.10.3. Cation-Exchange Capacity

The cation-exchange capacity was determined following the method of [23] with slight modifications. Briefly, an accurately weighed 1.0 g dried sample was placed in a 250 mL Erlenmeyer flask and mixed with 50 mL of 1 mol/L HCl, and kept at 25 °C for 24 h. The mixture was washed with distilled water to remove all the $Cl^-$ and then dried in an oven until a constant mass was obtained. The dried samples were then accurately weighed to 0.2 g and mixed with 50 mL of 5 g/100 mL NaCl, followed by the addition of two drops of phenolphthalein indicator. Distilled water was used as a blank sample, titrated with a 0.01 mol/L NaOH solution to the end point, and the volume of consumed NaOH solution was recorded. The cation-exchange capacity was then calculated as follows:

$$CEC(mmol/g) = \frac{(V_1 - V_0) \times C}{W} \tag{5}$$

where $V_1$ and $V_0$ are the volumes of NaOH solution consumed by titrating the sample and blank (mL), C is the concentration of NaOH solution used for titration (mol/L), and W is the mass of the dried sample (g).

### 2.11. Statistical Analysis

The average values obtained from three parallel samples were used as experimental data. Data were analyzed using SPSS software (SPSS for Windows, 22.0, 2012, SPSS Inc., USA) and Origin 8.0 (Origin Institute Inc, USA). Significant differences were determined using ANOVA test; a *p* value of less than 0.05 was considered statistically significant.

## 3. Results and Discussion

### 3.1. Structural Properties of RD-DF

### 3.1.1. SEM

The particle morphology of RB-DF at different processing stages was observed by SEM (Figure 1). D-RB showed a laminated fiber structure, with granular particles and smooth spheres that were likely composed of protein polymers and small starch granules. Following amylase and protease digestion, the number of granular particles and smooth spheres on the surface of RB-DF was significantly reduced, showing an irregular, large, dense, and compact laminated microstructure. Subsequently, cellulase hydrolysis led to changes in the RB-DF laminated structure (Figure 1B,C). This result is consistent with the study by [24], wherein cellulase treatment is shown to be beneficial for the formation of a rough and porous structure in DF. HPH treatment uses water as transmission medium, with its powerful mechanical force breaking the regular fiber structure and forming deep grooves [25]. Thus, the loose fiber structure with small particles and irregular flakes was attributed to the combination effect of cellulase hydrolysis and HPH treatment (Figure 1D).

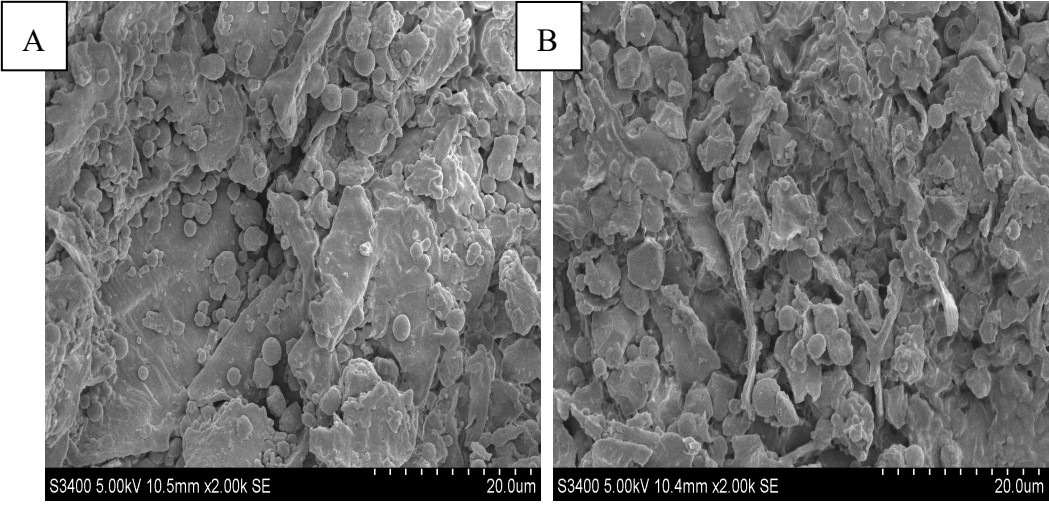

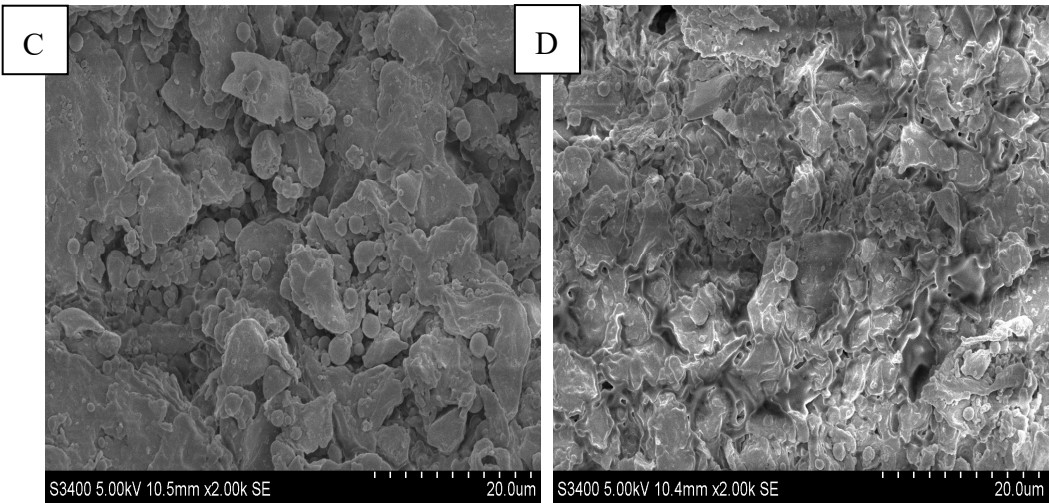

**Figure 1.** Scanning electron microscopy images of rice bran dietary fiber at different treatment stages (**A**) defatted rice bran, (**B**) rice bran dietary fiber, (**C**) cellulase enzymatic treatment rice bran dietary fiber, (**D**) high pressure treated rice bran dietary fiber.

### 3.1.2. Crystalline and Molecular Structure

The XRD patterns for D-RB, RB-DF, EH-DF, and HPH-DF (Figure 2) showed a main diffraction peak at 22.5°, which is attributed by cellulose and hemicellulose (2θ is 15–25°) [26]. However, diffraction peak intensity values do change significantly during sample processing ($p$ <0.05). The crystallinity of EH-DF decreased from 22.22% ± 1.23% to 11.76% ± 0.70% ($p$ <0.05) compared to that of raw D-RB. The main reason for this is the hydrolysis of starch and protein in rice bran, and the random and amorphous hemicellulose attached to cellulose microfibers surface is preferentially hydrolyzed during cellulase hydrolysis, resulting in a significant decrease in fiber crystallinity [27]. Subsequently, the crystallinity of HPH-DF increased from 11.76% ± 0.7%, to 15.78% ± 0.3%. Cellulose is a linear polymer of D-glucopyranose linked by β-1,4 glycosidic bonds, having a crystalline region and an amorphous region. The shear force generated by HPH likely promoted the cleavage of hydrogen bonds between cellulose molecular chains and destroyed the amorphous regions of cellulose. With the dissociation of the amorphous region, the arrangement regularity of cellulose molecule increased, resulting in an increase in crystallinity. Thus, HPH treatment leads to cellulose destruction of the connection between crystals, but does not affect the structural framework of the cellulose polymer backbone.

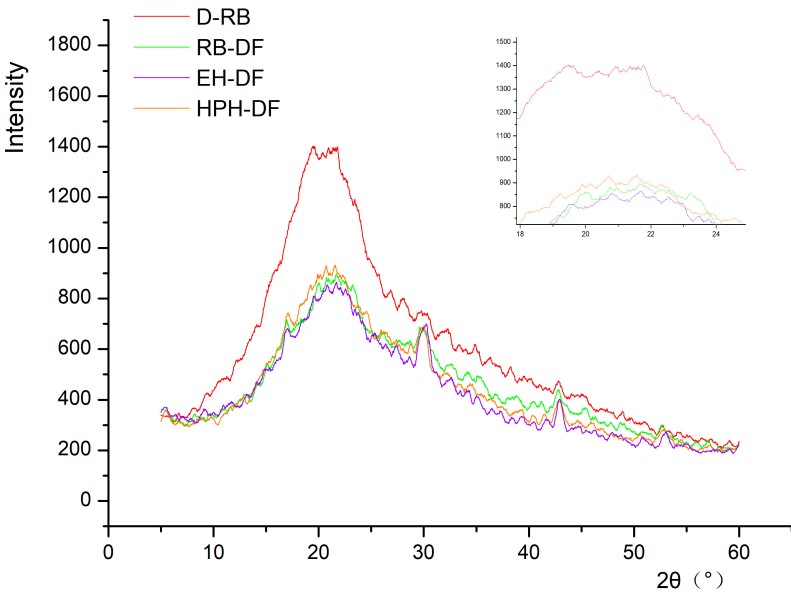

**Figure 2.** X-ray diffraction patterns of rice bran dietary fiber at different treatment stages. Inset shows main diffraction peaks about rice bran in four treatment stages. *D-RB* defatted rice bran, *EH-DF* cellulase enzymatic treatment rice bran dietary fiber, *HPH-DF* high-pressure treatment rice bran dietary fiber, *RB-DF* rice bran dietary fiber.

### 3.1.3. FTIR Spectra

In the infrared spectrum of RB-DF at various treatment stages (Figure 3), absorbance at 3323 cm$^{-1}$ and 2921 cm$^{-1}$ were assigned to hydroxyl groups and C-H bonds, respectively [28]. The absorption intensity of rice bran was decreased after being treated with amylase and protease, and the absorption intensity of RB-DF was increased after enzymatic hydrolysis and high-pressure treatment. This result indicates that the infrared absorption peak intensity of RB-DF is greatly reduced with the dissociation of starch from fat-removing D-RB. The widening and smoothing absorption peak of EH-DF and HPH-DF indicates that cellulase hydrolysis and HPH treatment exposed hydroxyl groups promotes disintegration of glycosidic bonds in DF structure [29]. Although the intensity of the characteristic absorption peak of carbonyl of cellulose uronic acid at 1743 cm$^{-1}$ was weakened, the characteristic absorption peaks of cellulose and oligosaccharide at 1647 cm$^{-1}$ and 1018 cm$^{-1}$ [30] were enhanced in EH-DF and HPH-DF. This phenomenon also indicates that cellulase hydrolysis and HPH treatment dissociate the non-crystalline regions in rice bran DF, promoting the degradation of small-molecule saccharides, and effectively changing RB-DF microstructure.

### 3.1.4. Thermal stability analysis

The differential TGA curves for RB-DF at various treatment stages were obtained from the first derivatives of weight loss rate (Figure 4). D-RB, RB-DF, EH-DF, and HPH-DF showed a weight loss peak (~3%) at 100 °C, attributed to the evaporated water. The TGA curves of the four samples had significant differences, whereas that for D-RB showed only one significant weight loss peak at 320–390 °C, and the corresponding weight loss weight was 42%, where 350 °C was its thermal decomposition temperature. For RB-DF, two weight loss peaks were observed at 290–360 °C and 380–500 °C, with thermal decomposition at 330 °C. The weight loss region for EH-DF was 278–397 °C, with thermal decomposition at 320 °C. Finally, for HPH-DF, the thermal decomposition temperature was 315 °C, with weight loss area at 260–380 °C corresponding to a 55% thermal weight loss rate, and a second weight loss area at 410–500 °C, where thermal loss weight loss rate reached 74%. An increase in weight loss rate is often accompanied by a decrease in thermal decomposition temperature [31,32]. In agreement with this, the weight loss rate of the four samples changed from 42–74%, and the thermal

decomposition temperature decreased from 315–350 °C. According to the abovementioned data, the thermal decomposition temperature and thermal stability of RB-DF decreased significantly ($p < 0.05$) following enzymatic hydrolysis and HPH treatment. It is possible that cellulase hydrolysis leads to cellulose particle loosening on the surface, with the subsequent high-pressure homogenization treatment reducing the particle size, and thus increasing the specific surface area. Additionally, a large number of active groups in RB-DF were exposed, increasing the surface adsorption capacity for low molecular weight fragments of cellulose, residual hydrogen ions, etc. These reactive groups, surface defects, and adsorbed residual ions led to RB-DF absorbing heat at a lower thermal decomposition temperature (315 °C vs. 330 °C) and to self-decompose.

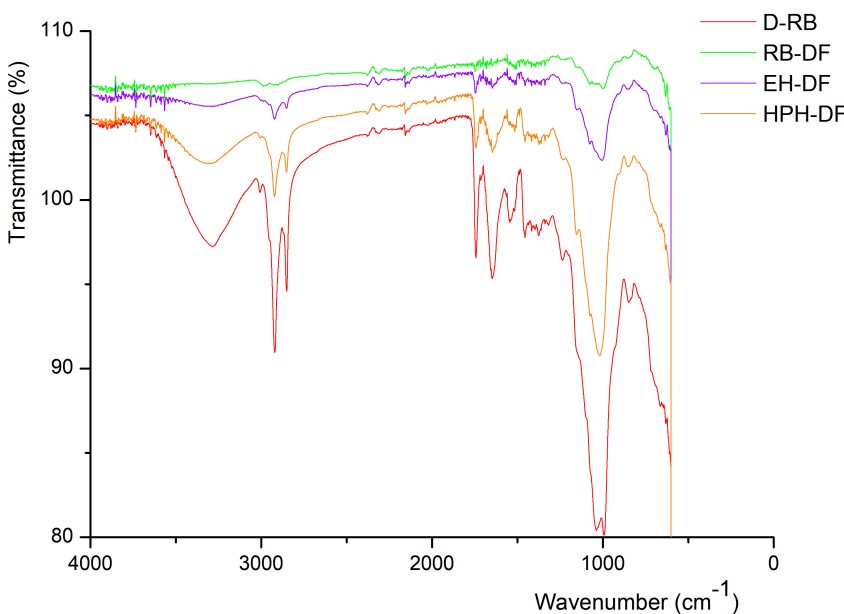

**Figure 3.** Infrared spectra of rice bran dietary fiber at different treatment stages. *D-RB* defatted rice bran, *EH-DF* cellulase enzymatic treatment rice bran dietary fiber, *HPH-DF* high-pressure treatment rice bran dietary fiber, *RB-DF* rice bran dietary fiber.

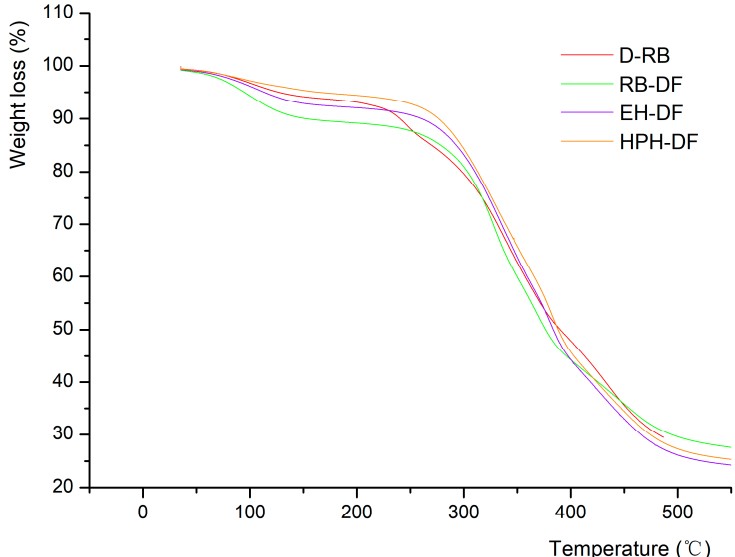

**Figure 4.** Differential thermogravimetric analysis curves of rice bran dietary fiber at different treatment stages. *D-RB* defatted rice bran, *EH-DF* cellulase enzymatic treatment rice bran dietary fiber, *HPH-DF* high-pressure treatment rice bran dietary fiber, *RB-DF* rice bran dietary fiber.

### 3.2. Physicochemical Properties of RB-DF

The physical and chemical properties of RB-DF at different stages of cellulase hydrolysis and HPH are shown in Table 1. With the removal of starch and protein by hydrolysis, RB-DF showed a significant change in oil and water holding capacity, expansion capacity, and cation-exchange capacity ($p$ <0.0.5). Following cellulase digestion and HPH treatment, the water holding capacity, expansion capacity, and cation exchange capacity of RB-DF continuously increased to 5.81 g/g, 5.26 mL/g, and 0.38 mmol/g, respectively, and the oil holding capacity decreased from 4.23 g/g to 3.43 g/g. Compared with cellulase, xylanase and ball-milling RD-DF [33], the cellulase and high pressure homogenization treatments have significantly improved the water holding capacity, swelling capability, and cation exchange capacity of RB-DF. The reason is that enzymatic hydrolysis and HPH treatment can change the microstructure of RB-DF, making it loose and porous, and exposing hydrophilic groups. Enhanced hydration resulted in an increase in water holding and expansion capacities. Therefore, it was inferred that a higher ability to retain water is related to a decrease in particle size of DF, due to the resultant breakdown of DF after being induced by high pressure treatment [34,35]. Additionally, the exposure of hydrophilic groups, such as carboxyl, hydroxyl, and amino groups, can produce a similar effect as anion exchange resins, allowing RB-DF to exchange cations such as $Ca^{2+}$, $Zn^{2+}$ and $Cu^{2+}$, maintaining stable pH and ion balance. Obviously, the chemical and structural natures of DF determine oil holding and swelling capacities. With the exposure of hydrophilic groups, the hydrophobic effect was weakened, reducing oil holding capacity. Therefore, the combination of cellulase hydrolysis and HPH treatment can micronize RB-DF, thereby improving the taste and nutrient absorption of DF foods.

**Table 1.** Physiochemical properties of rice bran dietary fiber at different treatment stages.

| Sample. | WHC (g/g) | OHC (g/g) | SC (mL/g) | CEC (mmol/g) |
|---------|-----------|-----------|-----------|--------------|
| D-RB | 3.02 ± 0.38 [a] | 3.13 ± 0.14 [a] | 1.00 ± 0.02 [a] | 0.21 ± 0.02 [a] |
| RB-DF | 3.20 ± 0.13 [ab] | 4.23 ± 0.13 [b] | 1.93 ± 0.09 [b] | 0.27 ± 0.01 [b] |
| EH-DF | 3.60 ± 0.27 [b] | 3.92 ± 0.11 [c] | 3.875 ± 0.01 [c] | 0.32 ± 0.01 [c] |
| HPH-DF | 5.81 ± 0.24 [c] | 3.43 ± 0.17 [d] | 5.26 ± 0.01 [d] | 0.38 ± 0.02 [d] |

Data were expressed by means ± standard deviation (SD). Values in the same column with different letters are significantly different ($p < 0.05$). CEC cation-exchange capacity, *D-RB* defatted rice bran, *EH-DF* cellulase enzymatic treatment rice bran dietary fiber, *HPH-DF* high-pressure treatment rice bran dietary fiber, *OHC* oil holding capacity, *RB-DF* rice bran dietary fiber, *SC* swelling capability, *WHC* water holding capacity.

## 4. Conclusion

Cellulase hydrolysis and HPH treatment are effective means for the modification of RB-DF structure. Compared with enzymatic treatment, the combined HPH and cellulase hydrolysis showed a significant effect on the degradation of RB-DF. Hydrolysis of hemicellulose and destruction of amorphous regions of cellulose can completely break the ordered laminar structure of DF, reducing its crystallinity and forming an irregular, small and loose particle structure. This results in a decrease in thermal stability, and a significant increase in water holding, expansion, and cation exchange capacities. This study helps clarify the underlying mechanisms of a combined biological and physical treatment for the structural modification of RB-DF, and provides valuable guidance for the micronization of DF.

**Author Contributions:** S.Z., Z.W., F.X. conceived and designed the experiments; F.X., T.Z., performed the experiments; H.W., M.L. and L.S. analyzed the data; F.X. wrote the paper.

**Funding:** This research was funded by the Natural Science Foundation of Heilongjiang Province grant number [C2018015/QC2016024], the Harbin Science and Technology Innovation Talents Research Project grant number [2017RAQXJ097], and Harbin Food Industry Research Institute Postdoctoral Innovation Entrepreneurship Practice Base.

**Conflicts of Interest:** The authors declare no conflict of interest.

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
