# Peer review of "Structural and Physicochemical Characteristics of Rice Bran Dietary Fiber by Cellulase and High-Pressure Homogenization"

_applsci, doi:10.3390/app9071270_

Round 1

Reviewer 1 Report

The work is clearly written in many parts. However, I felt the results could have been gone through more discussions. Please find a few suggestions

Line 27: The numbers do not provide much information here. Can be removed. 

Line 30: The context of grain size comes off suddenly at this point. 

Line 35: The current definition of dietary fiber also includes low molecular weight non-digestible polysaccharides. FYI

Line 62: The context of particle comes as a surpise. Suggested to indicate why small particle size is desired?

Line 71: What does setting of 4 for flow indicate in high pressure homogeniser?

In materials methods, do you have special reason to specify the concentration in ratios? The standard way is in %. Also for enzyme experiments, the standard way is to indicate the enzyme dosage in terms of enzyme activity in U or nkat/ g or mg of substrate.

In cation exchange capacity, can you please specify the reasons for modification of the method? For example, why HCl and NaCl was added?

Fig 2: Typo in Y-axis

Section 3.1.2 : This part could have been explained better

The crystallinity decreased upon the enzymatic treatment. The Line 170 says otherwise. On the other hand, the crystallinity is not very different with RB-DF.

Line 190: The context of water binding comes again off-topic here.

Author Response

The authors would like to thank the reviewer for useful feedback, and chose to upload a word file to response.

Reviewer 2 Report

Line 93: Provide more details from the freeze drying step; freezing temperature, gage pressure, and type of instrument used.

Lines 95-98: Provide details from the SEM analysis, don't just cite the reference.

The way Equation 1 is displayed is misleading. Provide separate equations for WHC and OHC (even if they are similar).

Units (such as g, mL, etc.) should not be expressed in Italics, even in the equations.

Line 165: There is an unnecessary question mark.

Lines 166-168: If you performed statistical analysis on the crystallinity and other properties, you should report standard deviations.

The FTIR interpretation is wrong. From the way the spectra are presented, authors measured Transmittance, not Absorbance. The removal of fat is not taken into account for interpretation. The lower Absorbance (or higher Transmittance) in the RB-DF sample is likely to be related to a substantial increase in porosity from the removal of fat. 

Author Response

(The authors gave the same response as above.)

Reviewer 3 Report

Authors have studied the effect of cellulase enzymatic hydrolysis and high-pressure homogenization on the some structural and some physicochemical properties of brown rice bran. Although the manuscript seems a bit poor there are some important outcomes regarding the structure modification of rice bran by cellulase enzymatic hydrolysis and HPH treatment.

See below some minor comments for revision:

1.     Authors should discuss also about cellulase enzymatic hydrolysis by ref. For example, they could provide information about other bran types in which enzymatic hydrolysis has been applied (Paz, Outeiriño, Pérez Guerra, & Domínguez, 2019; Zhao & Dong, 2016).

2.     Line 65: Authors should give information about the content of brown rice (ingredients).

3.     Line 74: Replace “were” with “was”.

4.     Lines 74-82: were these methods based in any previous publications? Please add reference.

5.     Lines 84-87 and 89-94: Please add reference regarding the used methods.

6.      Lines 97-98: Please add a brief description of the method. Were the samples coated with gold? Were the samples freeze-dried prior coating?

7.     Line 101: Please add a brief description of the method used. Do that in all materials and methods section.

8.     Line 150: Please revise reference style. Do that throughout the whole manuscript.

9.     Table 1: authors are advised to add more information in their Table comparing the results of other bran types with brown rice bran.

10.  Authors are advised to remove the keyword dietary fiber from the keywords as it is not an important representative of the present manuscript. Authors could add enzymatic hydrolysis or X-ray diffraction.

References

11.               Paz, A., Outeiriño, D., Pérez Guerra, N., & Domínguez, J. M. (2019). Enzymatic hydrolysis of brewer’s spent grain to obtain fermentable sugars. Bioresource Technology, 275, 402-409.

12.               Zhao, X., & Dong, C. (2016). Extracting xylooligosaccharides in wheat bran by screening and cellulase assisted enzymatic hydrolysis. International Journal of Biological Macromolecules, 92, 748-752.

Author Response

(The authors gave the same response as above.)

Round 2

Reviewer 2 Report

Your manuscript still needs correction if it is to be published. 

Freeze drying: If you operated the freeze dryer under vacuum, you could not have a gage pressure of 10 Pa.

In your FTIR results interpretation the only change you made was to add the suggestion I made in the first review. You need to support your results with previous works.

Author Response

Manuscript IDapplsci-464356

Title: Structural and physicochemical characteristics of rice bran dietary fiber by cellulase and high-pressure homogenization

Dear Editor,

Thank you so much for your letter and the reviewers’ comments, which are all very valuable and helpful for improving

our paper. We have revised the manuscript, and would like to re-submit it for your consideration.

We have addressed all the comments raised by the reviewers, and the amendments are highlighted in red and yellow

in the revised manuscript. Meanwhile, we tried our best to improve the manuscript and made some changes in the

manuscript, which will not influence the content and framework of the paper.

And here we did not list the changes but marked in red and yellow in revised paper.

Once again, thank you very much for your comments and suggestions. Point by point responses to the reviewers’

comments are listed below in this letter. If you have any question, please let me know.

We look forward to hearing from you.

With my best regards,

Zhong Jiang-Wang

Reviewer 3 Report

Authors have revised the manuscript as proposed. I suggest Accept in present form.

Author Response

Manuscript IDapplsci-464356

Title: Structural and physicochemical characteristics of rice bran dietary fiber by cellulase and high-pressure homogenization

Dear Editor,

Thank you so much for your letter and the reviewers’ comments, which are all very valuable and helpful for improving our paper. We have revised the manuscript, and would like to resubmit it for your consideration.

We have addressed all the comments raised by the reviewers, and the amendments are highlighted in red and yellow

in the revised manuscript. Meanwhile, we tried our best to improve the manuscript and made some changes in the

 manuscript, which will not influence the content and framework of the paper.

And here we did not list the changes but marked in red and yellow in revised paper.

Once again, thank you very much for your comments and suggestions. Point by point responses to the reviewers’

comments are listed below in this letter. If you have any question, please let me know.

We look forward to hearing from you.

With my best regards,

Zhong Jiang-Wang
